# What Do Butterflies Tell Us about an Intermediate Disturbance in a Dry Tropical Forest Context?

Yarlenis L. Mercado-Gómez [1], Jorge D. Mercado-Gómez [2] and Carlos E. Giraldo-Sánchez [3,*]

1   Grupo Evolución y Sistemática Tropical, Laboratorio de Conservación Biológica, Universidad de Sucre, Carrera 28 No. 5-267, Barrio Puerta Roja, Sincelejo 700001, Colombia; yarlenis.mercado@unisucre.edu.co
2   Grupo Evolución y Sistemática Tropical, Departamento de Biología y Química, Universidad de Sucre, Carrera 28 No. 5-267, Barrio Puerta Roja, Sincelejo 700001, Colombia; jorge.mercado@unisucre.edu.co
3   Grupo de Investigación de Sanidad Vegetal, Facultad de Ciencias Agropecuarias, Universidad Católica de Oriente, Sector 3, cra. 46 No. 40B 50, Rionegro 054048, Colombia
*   Correspondence: cegiral0@gmail.com or csanchez0@uco.edu.co

**Abstract:** Montes de María is the best-preserved tropical dry forest fragment in the Colombian Caribbean, making it an ideal location for studying the impacts of human disturbance on local ecosystems. In this study, we analyzed the community structure of diurnal butterflies in both forested and disturbed areas using 16 circular plots to identify relationships between alpha and beta diversity, and the geographic distance between disturbed areas and native forests, using a range of metrics, including range–abundance and rarefaction curves, nonlinear models, and the Bray–Curtis dissimilarity index. The results revealed three distinct species assemblages associated with forests, intermediate disturbed areas (IDAs), and disturbed areas (DAs). Nonlinear models show that IDAs are more diverse than forests and DAs. However, forests have more beta diversity in plots than IDAs and DAs. Indicator species for each butterfly assemblage were also identified. Thus, although new butterfly species assemblages emerge from a new human landscape, it is clear that species that only occur within dry forest fragments are lost when forest fragments disappear. Overall, these findings have important implications for conservation efforts and understanding how human disturbance affects biodiversity in tropical ecosystems.

**Keywords:** structure; diversity; anthropic disturbance; conservation; Lepidoptera

## 1. Introduction

A disturbance is, by definition, a discrete event in time that alters the ecosystem, community, population structure, resources, substrate availability, or physical environment [1]. The understanding of ecological patterns in disturbed communities is currently a central objective in ecology, primarily because it is known the disturbance affects the temporal or spatial variability of community structure (turnover and nesting; partitioned beta diversity), or both, as well as species richness (alpha diversity) [2]. Disturbances can cause convergence in community composition (lower beta diversity) by increasing suitable habitats for disturbance-tolerant species [3]. Alternatively, disturbance can lead to divergence in community composition (high beta diversity) by increasing the effect of the environmental filter on species composition [4,5].

One of the challenges in studying disturbance is understanding the mechanisms through which species diversity is maximized in a heterogeneous landscape [6]. Hypothetically, if areas closest to the forest are less disturbed, and the degree of disturbance increases with distance from the forest, intermediate areas should have moderate disturbance when compared to the two extremes of the disturbance gradient. In this sense, the intermediate disturbance hypothesis (IDH) is one of the most fundamental concepts. According to the IDH, there is a unimodal relationship between disturbance and diversity. An intermediate level of disturbance leads to higher levels of alpha biodiversity [7] due to compensations

between the ability of species to compete, colonize areas, and tolerate disturbance [8]. Although the IDH has influenced ecological theory, management, and conservation, its predictions are not always accurate across different taxa [9–11].

Communities can be structured taxonomically/phylogenetically when co-occurring species are highly or lowly phylogenetically related [12]. With the former, species communities shape the so-called phylogenetic clustering, and with the latter, they shape phylogenetic overdispersion [12]. Some studies in disturbed ecosystems have found phylogenic overdispersal in sites with intermediate disturbance [13,14]. Nonetheless, there is also evidence that the disturbance generates an environmental filter that only allows for the growth of phylogenetically related species, and therefore, a phylogenetic grouping is generated [15,16]. However, in addition to richness, the behavior of the phylogenetic structure also has a pattern that cannot be extended to all taxonomic groups, and therefore, the impact of disturbance on dry forest butterfly communities is little known [15].

Butterflies are an important model group to understand how human activities have influenced the community structure of tropical dry forest biota since they are highly sensitive to ecosystem changes [17–21]. Butterflies as model groups would be a strategy to generate information on the relationship of adjacent areas on the faunal diversity within dry forests without the need to explore all the biological groups, i.e., from an autoecological perspective [22–24]. In butterflies, e.g., diversity increases significantly as disturbance frequency increases [20,21]. Conversely, Addo-Fordjour, Osei and Kpontsu [17] and León-Cortés, Caballero, Miss-Barrera and Girón-Intzin [19] have found that the diversity of butterflies in disturbed habitats is considerably lower than in semi-natural preserved areas. Nevertheless, the mechanisms through which the disturbance influences the community structure are not generalizable since they depend on how the impact on the biota modifies environmental conditions and the colonization of species [5,21,25,26]. However, how disturbances affect the community structure of tropical dry forest butterflies is unknown.

Analyzing the effect of disturbance on the structure of the Lepidoptera Rophalocera butterfly communities of the tropical dry forest would provide a better understanding of how alpha and beta diversity indices are altered in areas with a disturbance gradient. In this sense, Montes de María constitutes a unique setting in the Colombian Caribbean to study how species assemblages change in environments modified by man in the tropical dry forest—mainly increasing—and thus uncover information that helps biodiversity preservation and restoration.

Neotropical seasonally dry forests (NSDFs) are one of the most biodiverse ecosystems in the world [27], and in the Americas, these are distributed discontinuously from Mexico to Argentina and throughout the Caribbean [28]. In the Neotropical region, these forests are highly threatened by human intervention [2], mainly due to logging, agriculture, and livestock, and suffer the effects of the expansion of urban areas [29]. In Colombia, these forests are distributed mainly over the inter-Andean valleys, with some isolated fragments toward the southern regions of the country in the Patía River Valley and, to a greater extent, in the Colombian Caribbean [29]. Currently, less than 4% of the original biome cover remains, and another 5% represents relics with some degree of anthropic disturbance. In other words, only less than 10% of NSDFs remain in Colombia [29]. In fact, of the 533,099 ha area that historically encompasses the dry forests of the Caribbean, only 37.97% still remains in a relatively good state of conservation, suggesting a strong anthropogenic disturbance impact on biotic communities [30,31].

Montes de María has been classified as one of the dry forest areas in the best state of conservation, mainly due to the implementation of the Protective Forest Reserve Serranía de Coraza and Montes de María [32]. However, agricultural activities have generated a mosaic landscape comprising patches of forest vegetation, pastures, and crops [31,33]. Therefore, human disturbance has modified the diversity and composition of butterflies. Nevertheless, how disturbance has structured the butterfly community is unknown. Herein, we analyze the effect of disturbance on the community structure of five Lepidoptera Rophalocera diurnal families (Papilionidae, Nymphalidae, Pieridae, Riodinidae, and Lycaenidae) currently

distributed in seasonally dry forests of the Coraza Reserve, Department of Sucre, Colombia. More specifically, we aimed to answer the following questions: How does species' alpha and beta diversity indices change according to the disturbance gradient in this area? Are there species assemblages that allow for the identification of areas with different degrees of disturbance? Has homogenization of beta diversity and taxonomic/phylogenetic structure occurred in the most disturbed communities? If these assemblages have been shaped by the degree of disturbance, are there indicator species for each of the assemblages?

## 2. Materials and Methods

### 2.1. Study Area

The Protected Forest Reserve Serranía de Coraza is located in the north of Colombia, in the Caribbean plains, in the Department of Sucre over the Montes de María [32] (Figure 1). Coraza shows temperatures ranging between 25 and 28 °C and precipitation between 896 and 1233 mm per year. Furthermore, there is a dry seasonal period from November to February where rainfall decreases to less than 200 mm. The average relative humidity is 83.5% [32,34].

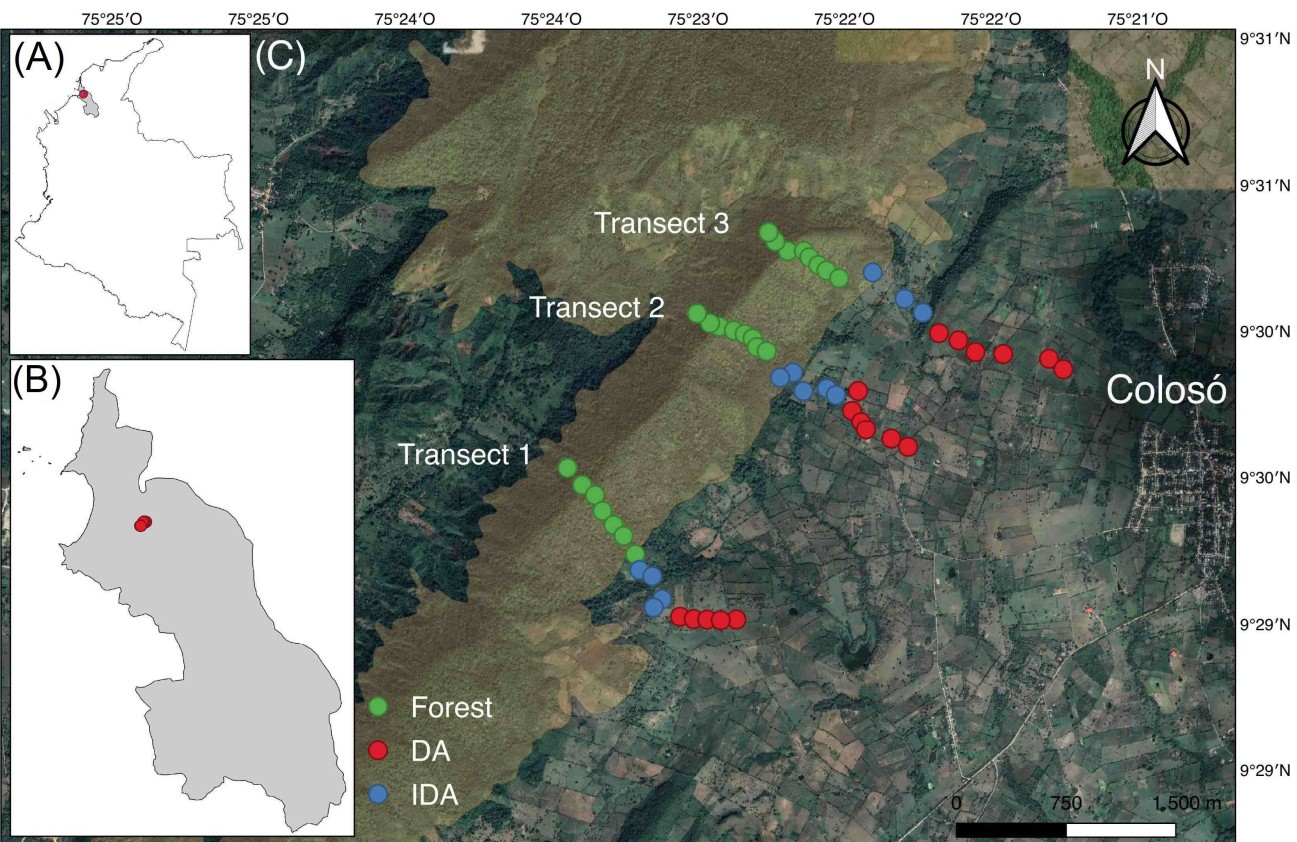

**Figure 1.** Study area: (**A**) Colombia, South America; (**B**) Department of Sucre; location of Protected Forest Reserve "Serranía de Coraza" in red; (**C**) linear transects and sampling plots.

The vegetation type belongs to a tropical dry forest with Fabaceae, Malvaceae, Meliaceae, Sapindaceae, Capparaceae, Rubiaceae, and Cactaceae as the most important plant families. The most abundant species are *Ampelocera edentula* Kuhlm (Ulmaceae), *Aspidosperma polyneuron* Mull. Arg (Apocynaceae), *Brosimum alicastrum* Swartz (Moraceae), *Myrcia fallax* (Rich.) DC (Myrtaceae), and *Simira cordifolia* (Hook. F) Steyerm (Rubiaceae) [35]. However, like much of the Colombian tropical dry forest, Montes de Maria forest fragments have been shaped into a mosaic of vegetation comprising forests, crops, and grazing-livestock areas [31].

## 2.2. Biological Material

In order to investigate the IDH on the butterfly communities of the tropical dry forest, we analyzed three linear transects 1200 m long. Each transect comprised sixteen circular plots; eight were located in the forest, and eight were in disturbed areas (pasture and crops). Each circular plot had a radius of 15 m, and the plots were separated from each other by a 60 m long distance [36]. A total of 48 circular plots (Figure 1) were sampled. Within each plot, butterflies were collected using an entomological net and Van Someren Rydon traps baited with a mixture of banana, rum, and sugarcane juice to attract frugivorous butterflies mainly of the Nymphalidae family [37]. Trap sampling was carried out during five effective hours per circular plot, located 2 m from the ground. Once the traps were installed, they were visited twice daily to collect the specimens. The collection with an entomological net was carried out between 8:00 and 16:00 h by three fieldwork personnel in each plot, with a total effort of 5 h of sampling per plot. We did not collect species over crepuscular times because of sociopolitical problems in the study area. We excluded the Hesperiidae family because it exhibits complexity and poses taxonomy problems in many genera, leading to the underestimation of their diversity in ecological studies and therefore ambiguity in the outcome of community structure studies [38–40].

The sampling was carried out during five field trips, lasting eight days each, between October and November 2014 and April and May 2015, where the highest rainfall peaks occurred [41], coinciding with the abundance of butterflies, according to Lucci Freitas, et al. [42]. Once the material was collected, it was stored in Milano paper envelopes [43] and transferred to the laboratory for later identification. Data regarding the plot number, locality (department and municipality), date, name of the collector, altitude, and geographic coordinates were collected for each specimen.

## 2.3. Taxonomical Identification

In order to carry out taxonomic identification, the collected specimens were rehydrated in humid chambers for a minimum of 24 h for their softening and subsequent assembly in the Biological Conservation Laboratory of Universidad de Sucre. Wing extension was performed following the protocol suggested by Triplehorn and Johnson [44]. Taxonomic identification was carried out mainly via comparison with photographs of type specimens deposited on the "Butterflies of America" website [45], field guides [46–48], and material deposited in the Entomological Museum of Universidad Nacional de Colombia, Medellín. For those individuals for which it was not possible to approximate a taxonomic entity through morphology, it was necessary to perform dissections of the genital organs using a liquid 10% KOH solution in a water bath for 15 min, for later observation in a stereoscope at 35X (Leika K100) and accurate taxonomic identification.

## 2.4. Community Structure Analyses

The expected number of species per cover type (forest and disturbed areas) was calculated using rarefaction and extrapolation–interpolation curves to establish whether the samples were representative of each transect. This method, described by Chao, et al. [49], uses the sample and a completeness curve drawn with twice the size of the smallest reference sample to be compared, with a 95% confidence interval obtained by resampling 100 bootstrap pseudoreplicates. This analysis was implemented with the R iNEXT package [50] following the parameters established by Chao, Gotelli, C, Elizabeth, MA, Colwell and Ellison [49] and Colwell, et al. [51].

To establish less disturbed and disturbed areas, we selected those sites that were analyzed by Iriarte-Cárdenas, et al. [52], who selected areas based on the descriptions of the CORINE Land Cover methodology adapted for Colombia. This methodology was used to identify areas that were less disturbed or preserved. Once located, the preserved areas are called watersheds (lines drawn on the highest peaks or pinnacles that enclose a basin area [53]). In fact, these areas are the least accessible areas for cultivation and livestock. We used Google Earth® to measure the distance between the circular plots located in

undisturbed areas and those located in disturbed areas and therefore identify areas with intermediate disturbance.

The above data were used to evaluate the change in richness and abundance associated with the distance from each plot to the least disturbed area of its transect (watershed), using linear and nonlinear regression models (linear, quadratic, potential, exponential, logistic, and Gaussian). The models' fit was assessed using the Akaike Information Criterion (AIC) [54]. The Gaussian model was the best-fit model, with values closest to zero. These outcomes show a relationship between species richness (Figure 2A), abundance (Figure 2B), and geographic distance. We found nine geographic ranges from the watershed (o m) to the most disturbed areas (>2050), where circular plots had similar values of species richness and abundance (0–250 m, 251–500 m, 501–750 m, 751–1000 m, 1001–1250 m, 1251–1500 m, 1501–1750 m, 1800–2050 m, and >2050).

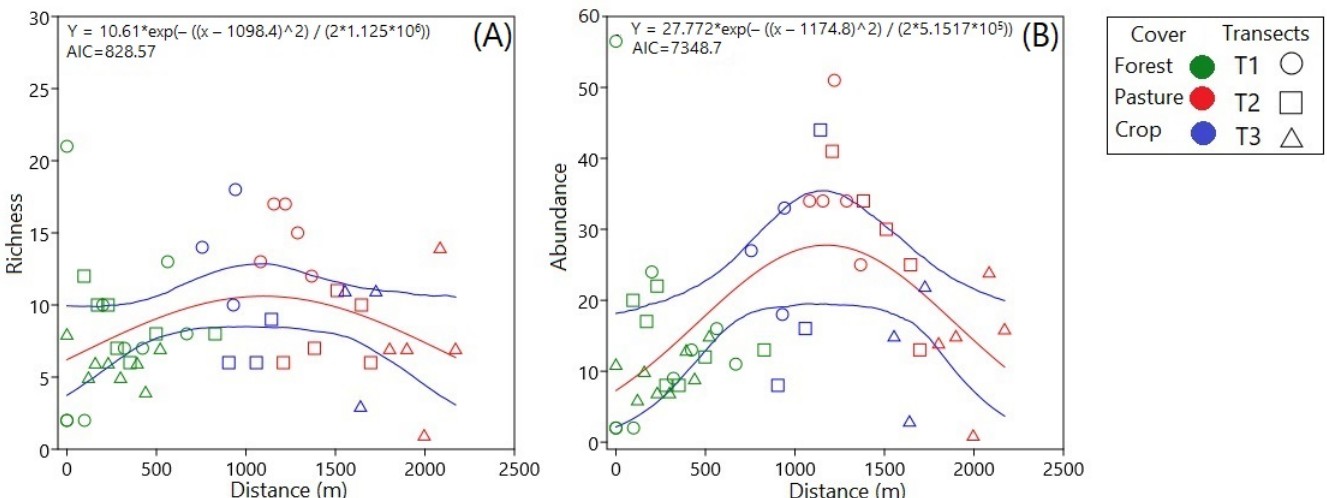

**Figure 2.** Relationship between the distance to the watersheds and the diversity of diurnal butterflies in the Serranía de Coraza, determined using a Gaussian nonlinear regression model, red line; 95% confidence intervals, blue lines: (**A**) richness; (**B**) abundance.

We built a new dataset of species abundance and geographic ranges. This matrix was employed to establish species assemblages using different measures of beta diversity [55]. Using the betapart package for R [56], we analyzed the extent to which species assemblages in the study area were shaped by spatial turnover versus nestedness. We employed the 'beta-pair.abund' function, which computes three distance matrices by using pairwise dissimilarities ($\beta$BC.BAL, $\beta$BC.GRA, and $\beta$BC).

The overall dissimilarity matrix, measured as Bray–Curtis ($\beta$BC) multi-site dissimilarity (average), was used to carry out a cluster analysis in the Vegan package for R [57]. The results from the dissimilarity matrices from Bray–Curtis revealed three species assemblages: (i) 0–750 m, (ii) 751–1750 m, and (iii) 1800–2050 m. The assemblages were categorized as forest (F), intermediate disturbed area (IDA), and disturbed area (DA). Then, the function 'beta.multi.abund' was employed, which uses the Bray–Curtis ($\beta$BC) dissimilarity index [58]. Its component of balanced variation ($\beta$BC.BAL) allows for the determination of whether communities are the result of turnover. This function also contains the component abundance gradient ($\beta$BC.GRA), which is used to identify nestedness [58].

Subsequently, a new species dataset was built between species abundance and the species assemblages (F, IDA, and DA). This new dataset was employed to analyze whether the homogenization of beta diversity has occurred in areas with higher disturbance using the effective number of communities ($^\beta$D) and the Whitaker beta diversity index ($\beta_w$). $^\beta$D was calculated using the gamma multiplicative partition D$\beta$ = D$\gamma$/D$\alpha$ (Jost 2007). We divided diversity into $\alpha$, $\beta$, and $\gamma$ components and computed the confidence interval for $\alpha$, $\beta$, and $\gamma$ using Monte Carlo simulations, assuming the species distribution among

plots and resampling them [59]. $^{\beta}$D has a minimum value of one, indicating no difference between sampling units (homogenization), and a maximum value equivalent to the number of sampling units when these do not share any species or heterogenization (Halffter and Ros 2013). The $\beta_w$ was calculated as β= γα in the PAST program (version 4.11) developed by Hammer, et al. [60]. Lower values of $^{\beta}$D suggest homogenization and higher values indicate heterogenization.

In addition, homogenization was analyzed through the taxonomic/phylogenetic community structure [61]. Warwick and Clarke [62] proposed an index to measure divergence and phylogenetic regularity [63]. The taxonomic distinctiveness index (Δ*) represents the taxonomic relationships between the species of an assemblage without considering richness or abundance [62]. The taxonomic distinctiveness variation index (Λ+) allows for the detection of the taxonomic equitability of assemblages, i.e., it measures the degree to which specific taxa are over- or under-represented in the samples [61,64]. The taxonomic indices were calculated with the PRIMER program (version 6).

Finally, an indicator species analysis (IndVal%) was carried out to establish whether there are indicator species in each assemblage. This methodology is key to identifying those taxa associated with greater weight in the different units that comprise the landscape, which is of higher relevance for species identified in forests or undisturbed areas used as a reference ecosystem for the analysis [65]. The species considered typical (indicators) of a habitat condition were those with InVal ≥ 50% [61]. These analyses were performed in PAST (version 4.11) [60].

## 3. Results

### 3.1. Taxonomic Composition

In total, 890 individuals from 5 families were collected, belonging to 15 subfamilies, 27 tribes, 63 genera, and 80 species (Table 1). The most representative family is Nymphalidae, with 605 individuals, representing 67% of the catches, followed by Pieridae (23%), Papilionidae (7%), Riodinidae (2%), and Lycaenidae (1%). Likewise, the richness of the families is in the following order: Nymphalidae (62%), Pieridae (17%), Papilionidae (9%), Riodinidae (5%), and Lycaenidae (5%). The subfamilies with the highest abundance were Biblidinae (272 individuals), Nymphalinae (159 individuals), and Coliadinae (138 individuals).

The most abundant species in forest areas are *H. februa* (56 individuals), *Junonia* sp1 (33 individuals), *Myscelia leucocyana* (C. Felder & R. Felder1861) (27 individuals), *Nica flavilla* (Godart, 1824) (18 individuals), and *Zaretis ellops* (Menetries, 1855) (16 individuals), which represent 17% of the total number of individuals collected in the study. On the other hand, in the disturbed areas (pasture and crops), 11 species were recorded as the most abundant. The most representative are *J. evarete* (111 individuals), *Eurema daira* (62 individuals), *Mestra hersilia* (Fabricius, 1776) (60 individuals), *Hamadryas feronia* (Linnaeus, 1758) (35 individuals), and *Ascia monuste* (Linnaeus, 1764) (30 individuals).

### 3.2. Ecological Structure of Communities

The species completeness of the study area was 89.04%, suggesting that the sampling was representative, reaching 93.7% for forests and 97.1% for disturbed areas (Figure 3). For each type of cover, sampling completeness estimates slightly differed by increasing the size of the reference sample in terms of individuals to double (from 302 to 604 in forest and 600 to 1200 in disturbed areas). This means that, even if more individuals were collected in the study area, the sampling would continue to represent both the forest and disturbed areas (Figure 3A). When comparing the species richness concerning the number of individuals between the types of coverage (forests and disturbed areas), it was possible to establish, with a confidence interval of 95%, that the expected richness of butterflies would continue to be higher in disturbed areas than in forests even when all extant species were recorded (Figure 3B).

**Table 1.** List of diurnal butterfly species identified in the Coraza Reserve in three types of cover: forest, pasture, and crops.

| | | |
|---|---|---|
| *Adelpha fessonia ernestoi* [1] | *Euptoieta hegesia* [3] | *Mesosemia carissima* [1] |
| *Adelpha iphicleola* [1] | *Eurema agave* [3] | *Mestra hersilia* [3] |
| *Agraulis vanillae* [3] | *Eurema arbela* [2] | *Microtia elva* [3] |
| *Anartia amathea* [3] | *Eurema daira* [2] | *Morpho helenor* [1] |
| *Anartia jatrophae* [3] | *Eurema elathea* [3] | *Myscelia leucocyana* [1] |
| *Anteos maerula* [3] | *Fountainea halice* [2] | *Neographium anaxilaus* [1] |
| *Archaeoprepona demophon* [1] | *Glutophrissa drusilla* [1] | *Nica flavilla* [1] |
| *Archaeoprepona demophoon* [1] | *Hamadryas februa* [2] | *Parides anchises serapis* [2] |
| *Aricoris erostratus* [3] | *Hamadryas feronia* [2] | *Parides eurimedes mycale* [1] |
| *Ascia monuste* [2] | *Heliconius erato* [2] | *Parides iphidamas* [2] |
| *Battus polydamas* [2] | *Heliconius ethilla* [1] | *Phoebis agarithe* [3] |
| *Biblis hyperia* [3] | *Heraclides thoasnealces* [3] | *Phoebis argante* [2] |
| *Caligo brasiliensis morpheus* [1] | *Hermeuptychia hermes* [1] | *Phoebis sennae* [3] |
| *Callicore pitheas* [1] | *Historis odius* [3] | *Prepona laertes* [2] |
| *Chlosyne lacinia* [2] | *Hypna clytemnestra* [2] | *Pseudolycaena marsyas* [3] |
| *Chlosyne poecile* [2] | *Itaballia demophile* [2] | *Pyrisitia dina* [1] |
| *Cissia themis* [2] | *Itaballia pandosia* [1] | *Pyrisitia leuce* [1] |
| *Colobura dirce* [1] | *Janatella leucodesma* [1] | *Pyrrhogyra neaerea* [1] |
| *Consul fabius* [1] | *Juditha* sp. [3] | *Siderone galanthis* [2] |
| *Danaus eresimus* [3] | *Junonia* sp1 [3] | *Siproeta stelenes* [1] |
| *Danaus gilippus* [2] | *Junonia* sp2 [3] | *Smyrna blomfildia* [1] |
| *Detritivora hermodora* [2] | *Leptotes cassius* [3] | *Strymon* sp. [3] |
| *Doxocopa pavon theodora* [2] | *Libytheana carinenta* [1] | *Taygetis laches* [1] |
| *Dryadula phaetusa* [3] | *Lycorea halia* [2] | *Temenis laothoe* [2] |
| *Dryas iulia* [2] | *Marpesia chiron* [3] | *Thereus cithonius* [3] |
| *Ectima erycinoides* [1] | *Melanis electron* [1] | *Zaretis ellos* [1] |
| *Eunica tatila* [1] | *Memphis arginussa* [1] | |

[1] Exclusive to forests; [2] occurring in both forest and pasture–crop areas; [3] Exclusive to pasture–crop.

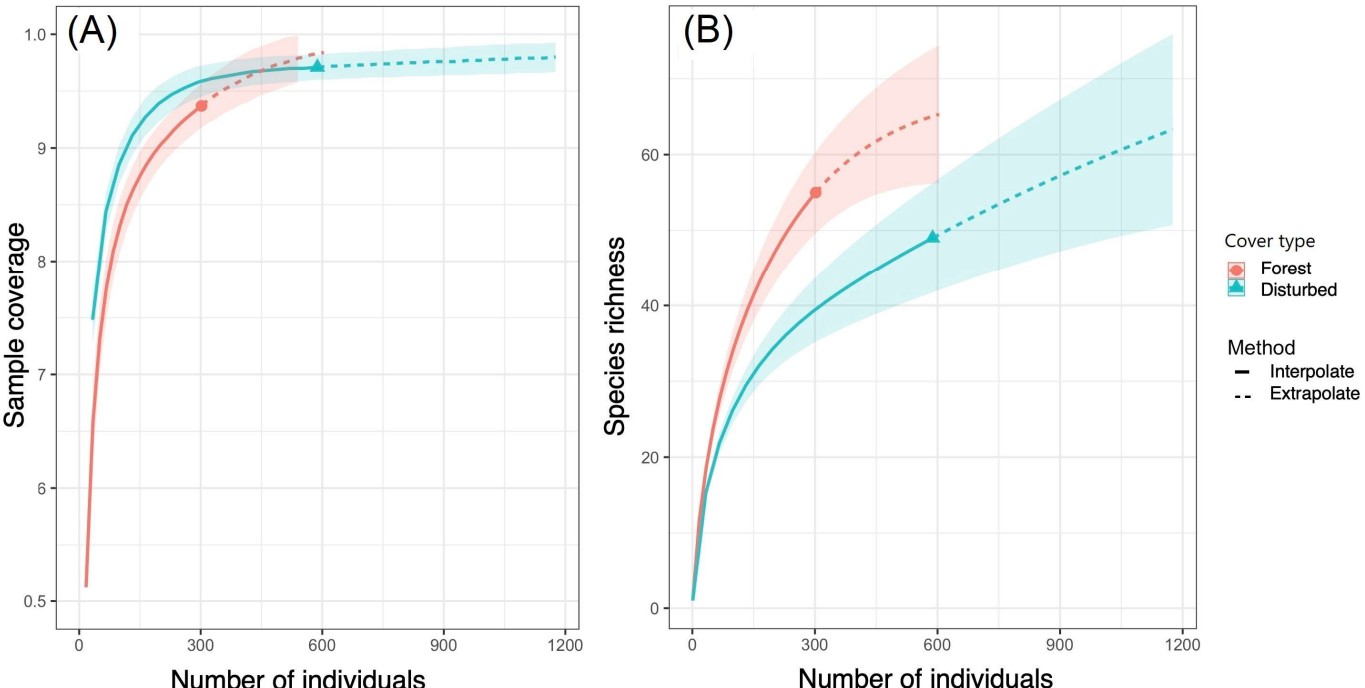

**Figure 3.** Rarefaction and interpolation–extrapolation curves, based on the diurnal butterfly community by cover type (forest or disturbed); solid lines represent the estimation using interpolation, and dashed lines represent extrapolation: (**A**) sampling completeness; (**B**) richness estimation.

The cluster analysis results based on Bray–Curtis allowed us to distinguish three large groups: All plots located between 250 and 750 m correspond to the forest area, those between 751 and 1500 m correspond to IDA, and those between 1800 and 2050 correspond to DA (Figure 4). In the case of nesting and species turnover results, βBRY.GRA = 0.027 and βBRY.BAL = 0.937. The beta diversity analyses show a general pattern in which the forest area records higher values ($^{\beta}$D = 1.63; $\beta_w$ = 0.94). By contrast, IDA registers lower values than the forest area ($^{\beta}$D = 1.51; $\beta_w$ = 0.61), and DA has lower values than IDA ($^{\beta}$D = 1.48; $\beta_w$ = 0.60).

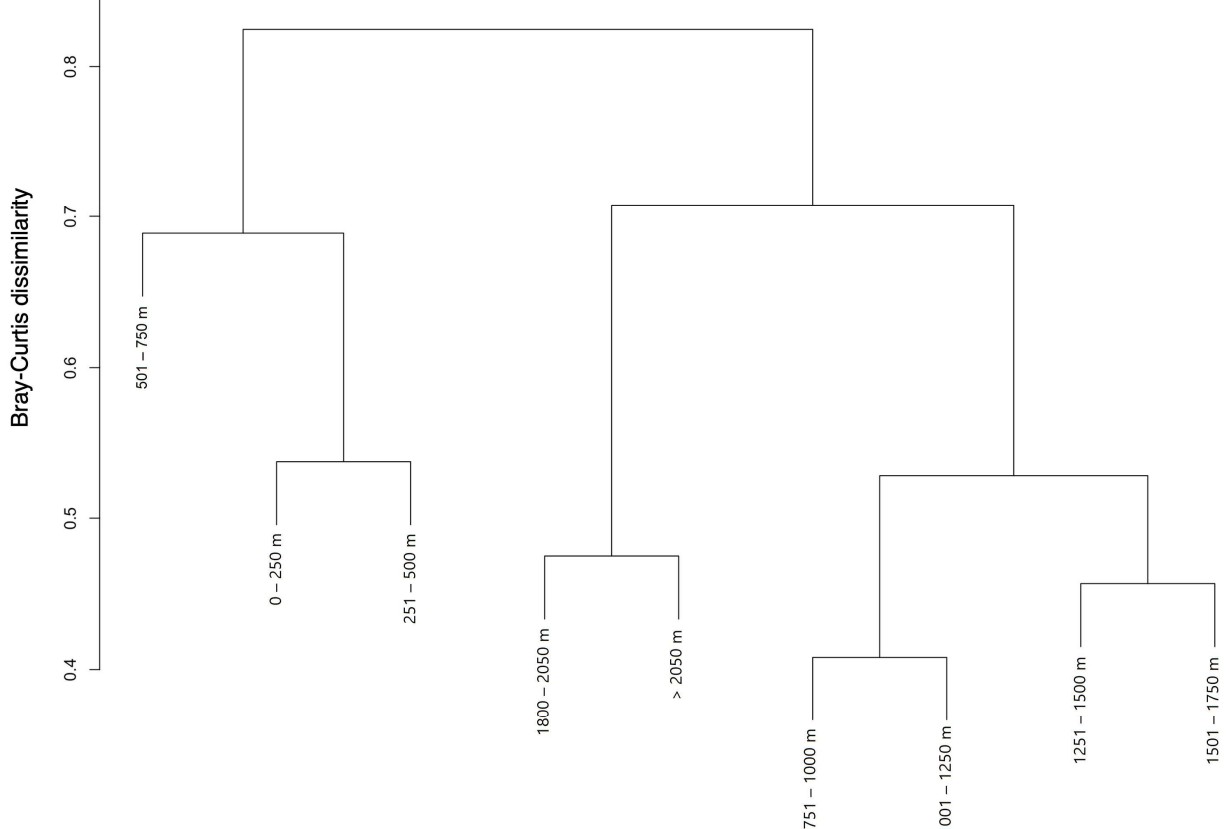

**Figure 4.** Cluster analysis of the sampling plots, based on Bray–Curtis distance, grouped according to the distance to watersheds.

The taxonomic distinctiveness model (Δ*) and its variation (Λ+) built at the level of the analyzed communities (F, IDA, and DA) show that they are within the 95% confidence intervals. In this sense, Δ* was 49.56 for F, 52.26 for IDA, and 52.11 for DA. On the other hand, the variation values in taxonomic distinctiveness (Λ+) were 92.41 for F, 102.51 for IDA, and 90.30 for DA.

A total of 21 specific species were found for B (undisturbed), IDA, and DA assemblages through the InVal analysis (Figure 5). In the forest area, a total of nine species were found. *Archaeoprepona demophon*, *Hamadryas februa*, *Itaballia demophile c*, *Morpho helenor*, *Myscelia leucocyana*, *Nica flavilla*, *Siderone galanthis*, and *Zaretis ellops* are typical of this cover type (InVal ≥ 50%, *p* = 0.001). In the case of areas with intermediate disturbance, with values of InVal ≥ 50% and *p* = 0.001, *Biblis hyperia*, *Eurema daira*, *Hamadryas feronia*, Junonia sp2, *Mestra hersilia*, *Phoebis sennae*, and *Temenis laothoe* are specific taxa of these areas. Finally, in disturbed areas, only *Battus polydamas*, *Danaus eresimus*, *Danaus gilippus*, and *Dryadula phaetusa* are typical species of these areas (InVal ≥ 60% and *p* = 0.001).

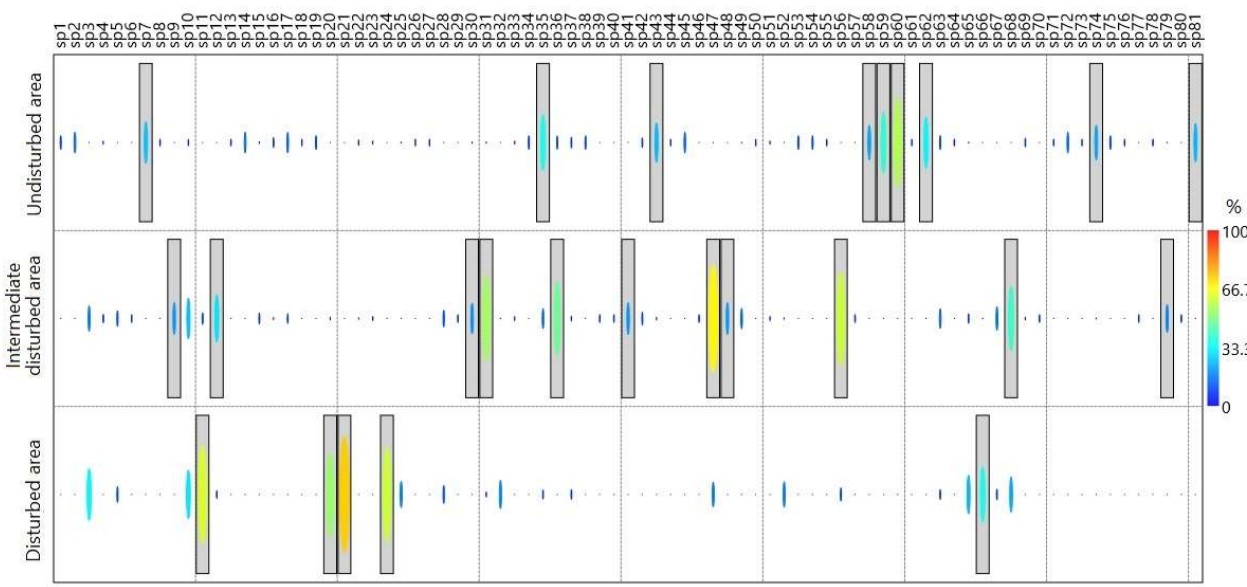

**Figure 5.** Indicator species analysis (IndVal%) for undisturbed areas or forests, areas with intermediate disturbance, and disturbed areas. Species with values above 70% (reddish tones) are considered indicators; gray boxes denote species for which the indicator values have significance (*p* value < 0.05).

## 4. Discussion

### 4.1. Taxonomic Composition

Th composition outcomes show that the diversity of Coraza is similar to those found in other tropical dry forest fragments in Colombia [66]. Nymphalidae was the most representative family regarding the number of individuals and species recorded in most studies in Colombia. At the Patía River Basin, Millan, et al. [67], Gaviria-Ortiz and Henao-Bañol [68], and Henao-Bañol and Andrade-C [69] registered 60 and 90 species, respectively, while Orozco, et al. [70] identified 78. In the Magdalena River Valley, Peña and Reinoso [71] found 64 species, and Prince-Chacon, et al. [72] reported 38 in the Caribbean plains. Lamas (2004) pointed out that Riodinidae have the highest richness after Nymphalidae. However, besides Riodinidae, species from the Lycaenidae family have also been reported with the lowest richness in other studies of butterflies in the tropical dry forest [70,73], mainly due to their small size and dark colors [74]. In addition, most of the species of these families tend to fly in higher strata and rest in the upper part of the trees, making their capture in forest areas difficult [75].

Biblidinae is the most abundant subfamily and has the highest richness. Regarding studies on the richness of this subfamily in dry forest fragments in the Caribbean region, Montero, et al. [76] and Vargas Zapata, et al. [77] found their species to have the highest values with 13 and 10 taxa, respectively. Nevertheless, Vargas Zapata, et al. [77] found only six species in a dry forest fragment in the Department of Atlántico, in addition revealed that the highest richness of Biblidinae may be related to the ability of its species to use a wide variety of resources in different plant strata. The abundance results of Biblidinae may be related to the larval stage of species in this subfamily since they feed on plants belonging to the Malvaceae, Euphorbiaceae, Moraceae, and Sapindaceae families [78], typical of these dry ecosystems. Herazo-Vitola, Mendoza-Cifuentes and Mercado-Gómez [35] indicated that the Malvaceae, Sapindaceae, and Euphorbiaceae are of great relevance in terms of richness and dominance in the flora of Montes de María.

### 4.2. Community Structure Analysis

We found that forest disturbance in the tropical dry forest of Montes de María due to the expansion of the agricultural frontier (crops and pastures) has led to species loss, thus changing the dynamic of the butterfly community structure. The regression outcomes and

the cluster analyses based on the dissimilarity matrices allowed for the identification of species assemblages correlated with the geographical distance (m). Preserved forest areas and disturbed areas (crops and pastures) were found. Three zones were highlighted in the present study: undisturbed forest areas, intermediate disturbed areas, and disturbed areas. The regressions show that forest areas are characterized by low diversity, increasing in intermediate disturbed areas and decreasing in disturbed areas.

These data support the intermediate disturbance hypothesis proposed by Connell [79], according to which an increase in diversity is observed in intermediate areas. This pattern in diversity has also been found in humid forests, where butterfly diversity values are higher in an intermediate disturbed area than in a disturbed forest [17,19,80]. The formation of these species assemblages may be the result of a long history of anthropic transformation that the Coraza Reserve has endured in Montes de María. When forest fragments are cleared for cattle grazing or crop cultivation, the habitat for butterflies disappears, leading to the dispersion of species to nearby fragments.

Increases in species richness are often due to the invasion of disturbed areas by generalist and widely distributed species [18]. According to Vanschoenwinkel, et al. [81], diversity increases because many forest species are more tolerant to disturbance than expected by chance. Thus, extinction rates mediated by stochastic events such as crop and pasture implementation are not necessarily deterministic (i.e., species have similar extinction probabilities). In other words, disturbance can promote alpha diversity under these conditions.

Disturbance was also found to affect another dimension of butterfly communities' structure. Beta diversity decreases as disturbance increases. The forest area has higher beta diversity than IDA and DA; however, IDA is higher than DA. The diversity homogenization process in the most disturbed communities may result from the ecological filter [82]. Environmental conditions, e.g., higher temperatures and solar radiation in disturbed areas compared with forest areas, can create barriers that prevent not only movement between forest species and disturbed areas but also the formation of communities with some functional traits that increase their tolerance toward more environmentally stressful areas such as pastures and crops [83,84]. However, these aspects must be evaluated in depth, where the functional traits of butterflies and an analysis of their phylogenetic structure can be measured.

The divergence and phylogenetic regularity analysis measured through the taxonomic distinctiveness of the three types of assemblages show no apparent differences among IDA, forest, and DA. However, IDA is slightly more diverse than DA and forest assemblages. These outcomes suggest that species of IDA have a slightly higher phylogenetic separation between species or greater evolutionary distances between their taxa than the rest of the assemblages [85]. Nevertheless, although the IDA assemblage has a higher species richness, the distribution of its species in the higher taxonomic categories (phylogenetic divergence) has a value equivalent to that of the DA assemblage [85].

Conversely, the low values in the forest area indicate that the species are more closely related at lower taxonomic levels (e.g., several species belonging to the same genus or family). Therefore, these species are more phylogenetically related than those in disturbed areas. This study proposes two arguments to explain the high phylogenetic relationship or phylogenetic grouping in dry forest butterfly communities [86,87]. The first is based on ecological forces such as the ecological or environmental filter, indicating that the environment is a filter that allows only species with particular traits or phenotypes to establish and persist in the forest [88]. The second is more evolutive, based on phylogenetic niche conservatism, which indicates that closely related species are more similar in ecological, morphological, and functional traits than distantly related species since they have inherited it from their ancestors [89].

On the other hand, outcomes from IDA and DA show a community with higher phylogenetic overdispersion, i.e., a higher number of less taxonomically related taxa. Webb, et al. [90] suggest that overdispersion in communities is mainly the product of

competitive exclusion. Most taxonomically related species have a higher morphological similarity in their behavior and eating habits. Similar species tend to compete for the same resources when exposed to a new niche. However, those that become the leading taxa more easily adapt to obtain resources, generating exclusion through competition with other phylogenetically related species [12].

In the case of the Λ+ results, a difference from Δ* is observed, as IDA and the forest area have higher values than DAs. The high values of Λ+ are related to an excessive or insufficient representation of some taxonomic groups (irregularities in the taxonomic tree). In IDA and forest, species are concentrated in the subfamilies Charaxinae, Coliadinae, Papilioninae, and Nymphalinae. The low Λ+ values in the disturbed areas indicate that the variance in the distribution of their species in the higher taxonomic categories is lower (lower Λ+ value), which can be reflected in the fact that their species are grouped mainly in Coliadinae. In other words, the supraspecific distribution is more equitable than the assemblages in intermediate disturbed and forest areas [85]. These results suggest that the species found in areas with intermediate disturbance have a degree of phylogenetic regularity similar to that of the forest. Therefore, ecological processes, such as competition, influence the under- or over-representation of taxa in their communities. However, in both cases, it is necessary to analyze functional traits to obtain a higher degree of precision on the effects of ecological or evolutionary processes on butterfly communities in forests and areas with the two levels of disturbance.

Disturbance affects butterfly communities in dry forests not only in terms of the generation of new communities but also in that the species in these assemblages have adapted to their new habitats to such an extent that their distribution is restricted to these new areas, so they no longer exist in forests. *Biblis hyperia*, *Eurema daira*, *Hamadryas feronia*, *Junonia* sp2, *Mestra hersilia*, *Phoebis sennae*, *Temenis laothoe*, *Battus polydamas*, *Danaus eresimus*, *Danaus gilippus*, and *Dryadula phaetusa* are exclusive species in IDA and DA. When a forest is disturbed, species of butterflies sensitive to disturbance tend to disappear, while more tolerant species persist [91]. This study suggests that the species of butterflies associated with IDA and DA can be considered tolerant taxa toward extreme anthropic events because they can efficiently take advantage of the resources the environment offers [92].

An indicative species of the forest area is *Archaeoprepona demophon* (Linnaeus, 1758), belonging to the subfamily Charaxinae. *A. demophon* prefers decomposing fruits, which is supported by the large number of individuals found in the baited traps. Forests in the study area have species of the Sapindaceae, Moraceae, Anacardiaceae, and Myrtaceae families, which produce sweet and edible fruits that decompose when falling to the ground, and butterflies can absorb their sugars [67]. *Morpho helenor* was found inside forest areas because it depends on little-disturbed stream edges and forest interiors [67] due to its complex habitat requirements. In addition, species such as *Hamadryas februa*, *Itaballia demophile*, *Myscelia leucocyana*, *Nica flavilla*, *Siderone galanthis*, and *Zaretis ellops* were also exclusive to forest areas. This can be explained by the availability of resources for adults, host plants for larvae, or the environmental humidity due to nearby water sources [70] offered by forests, unlike other cover types. An interesting case was *H. februa* because this species was also found in typically disturbed areas in previous studies. This may be because these studies were conducted in other biomes or habitat types [93].

All nine species of butterflies registered as exclusive to the forest were found to have fewer than three individuals. According to the scale proposed by Fagua [94], species with fewer than three individuals are considered rare, and their occurrence is related to discontinuous flight periods or alternate imago emergence events in different butterfly species. Master [95] proposed that the occurrence of rare species within forests may indicate areas of interest for conservation, suggesting the importance of preserving the dry ecosystems of the Protective Forest Reserve Serranía de Coraza.

## 5. Conclusions

Seasonally dry forests are currently considered one of the ecosystems with the highest degree of threat, mainly due to anthropic activities such as livestock grazing, cultivation, and urban expansion. These activities have been modifying the structure of biotic communities. Species diversity has been altered, with a reduction in richness and possible changes in species composition. However, hypotheses on the behavior of species communities, for instance, in areas with intermediate disturbance, have not been tested in this type of ecosystem in the Colombian Caribbean. In this sense, the results of the current study corroborate that anthropic disturbance has allowed for the formation of areas with intermediate disturbance, which show a higher alpha diversity, in terms of both taxonomic and phylogenetic structure.

In the same way, it is clear that anthropic activities have resulted in the formation of two new assemblages of butterfly species associated with intermediate disturbed and disturbed areas, in which homogenization has occurred in beta diversity, which is an outcome of a progressive decrease in diversity. Likewise, these new communities have also undergone phylogenetic homogenization where most of their species exhibit a higher taxonomic relationship than expected by chance. This may be the result of ecological processes such as competition or the ecological filter. However, it is necessary to further analyze functional traits and, thus, establish new hypotheses about the structure and the formation of the communities associated with disturbed areas.

In the same way, these new communities have species that are indicative of or exclusive to them since they are not found in other areas. This suggests that some forest taxa are not very sensitive to changes in land cover, but there are also species that are more malleable to environmental changes. According to the above explanation, the transformation of forest to agricultural land cover would imply local losses of species associated with tropical dry forests. This result is highly relevant, as butterflies can quickly generate new species assemblages according to environmental changes. It is also clear that species that only occur in a forest will be lost when cutting down the forest.

**Author Contributions:** Conceptualization, C.E.G.-S. and J.D.M.-G.; methodology, C.E.G.-S. and J.D.M.-G.; formal analysis, C.E.G.-S. and J.D.M.-G.; investigation, Y.L.M.-G., C.E.G.-S. and J.D.M.-G.; resources, Y.L.M.-G. and J.D.M.-G.; data curation, Y.L.M.-G. and C.E.G.-S.; writing—original draft preparation, J.D.M.-G.; writing—review and editing, J.D.M.-G. and C.E.G.-S. All authors have read and agreed to the published version of the manuscript.

**Funding:** This research received no external funding.

**Institutional Review Board Statement:** On behalf of Universidad de Sucre, ANLA approved the biological collection of butterfly specimens in the study area (permit number 0391 of 2016).

**Data Availability Statement:** Data are available from the corresponding author upon request.

**Acknowledgments:** We would like to thank Pedro Álvarez, Daniel Peralta, and Luis Andrés Severiche for their assistance in collecting the butterflies and tracing the transects in the field. We would also like to thank Universidad de Sucre for lending its facilities for research development.

**Conflicts of Interest:** The authors declare no conflict of interest.

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
