# Peer review of "What Do Butterflies Tell Us about an Intermediate Disturbance in a Dry Tropical Forest Context?"

_diversity, doi:10.3390/d15080927_

Round 1

Reviewer 1 Report

By comparing the butterfly communities of neotropical seasonally dry forests in Colombia with those of the adjacent human-disturbed areas, the authors clarified the changing patterns in butterfly species diversity and community structure along the gradient of anthropogenic disturbance and how anthropogenic disturbance affects and changes the butterfly communities of the neotropical seasonal dry forest. Thus, the results of this study are considered to be appropriate for publication in this journal.

On the other hand, the results and community patterns identified in this study have already been described in many previous studies and are not highly novel. But, most of the known community patterns have been identified in wet and humid tropical forests, and in this regard, it is highly commendable that the known community patterns were found in the butterfly community of the neotropical seasonally dry forest that was the subject of this study. In addition, the originality of the butterfly community data obtained in this study is very high and well worth publication.

Overall, the paper is very well written and well argued, with no major revisions requested. A few minor comments I noticed are noted below.

Lines 138-142: Are there butterflies in the targeted butterfly communities that are not attracted to the traps (e.g., saprophyte specialists, flower nectar specialists, etc.)? If so, it is necessary to discuss whether there is any impact on the butterfly community analysis.

Line 241: The explanatory text in Table 1 is in Spanish? It should be in English.

Lines 248-256: The butterfly community data in the forest and disturbed areas are described, but no data on IDA's butterfly community is presented. Is there a reason for this?

Author Response

Please see the attachment. Thank you for providing comprehensive feedback on our manuscript. 

Reviewer 2 Report

 General comments:

A complete revision of the English will be necessary; some parts are extremely difficult to understand. This article looks interesting but the current version is clearly not acceptable as it stands. Also, please present your results graphically, otherwise your manuscript is difficult and unattractive to read. Please also include a map of your study site and transects. I like this paper and it is certain that due to the quantity and quality of the fieldwork (although Junonia will need to be checked) it can be published, but there is still a lot of work to be done. Please include (in the supplementary material) all your field data with occurrences in the different areas. The conclusions should be modulated (see my comments at the end). With all due respect, your current manuscript seems to be an advanced draft of a much more elaborate future document.

Detailed corrections:

The title is not impactful enough and is too long. Something like this might be better: "What do butterflies tell us about an intermediate disturbance in a dry tropical forest context?"

The location of the study is not important in the title, as the main focus should be the response to this type/level of disturbance.

Abstract:

Line 19: What do you mean by ecological structure? I suppose that it is more complexity/dynamic/diversity of community?

Introduction:

First paragraph is not there (line 37 to 49). You’ve got to start by your general question in ecology before speaking of your local context.

Your introduction is starting line 50.

Line 54: Definition of beta diversity is not correct or at least incomplete. You must include the word comparison.

Reading your text. I think that you’ve got to restructure it.

Start by line 50 to 68, then line 83 to 92, then line 78 to 82 (in this section some literature is missing, please check literature of Pozo Carmen, Janzen Daniel and Legal Luc), then line 70 to 77, then to combine with line 92 to 99 and only then presenting your site line 37 to 49 combined with line 100 to 109. Finally please separate in a paragraph your aims from line 109 to line 115.

Material and methods:

Line 126: please try to combine species names with their respective family.

I fully understand the potential risk in such an area of social instability. Anyway, why not collect until 18:00? As you probably know, many Riodinidae (or if you prefer Riodininae) start to be active around 15:30 and until 18:00 in this type of forest.

Line 144: Line 154: Remove names. This reference is not correct in your text and bibliography please check it (see below Freitas is the “apellido” and André Victor Lucci the “nombres”). Your dates are wrong because André's team (Freitas) mentions collections in southern Brazil and I did not find any mention of these rainfall peaks in their article. After checking, I found the source of your text: https://www.colombia.co/en/colombia-travel/faq-climate-weather-colombia/

Please try to get more scientific weather data.

Freitas, André Victor Lucci, Agra Iserhard, Cristiano, Pereira Santos, Jessie, Oliveira CarreiraI, Junia Yasmin, Bandini Ribeiro, Danilo, Alves Melo, Douglas Henrique, Batista Rosa, Augusto Henrique, Marini-filho, Onildo João, Mattos Accacio, Gustavo, & Uehara-prado, Marcio. (2014). Studies with butterfly bait traps: an overview. Revista Colombiana de Entomología, 40(2), 203-212.

Line 182: Once again, remove names. Please check which Chao index you were using Chao1 or Chao2.

Line 215: I think that in your case it could be interesting to apply this approach for beta diversity calculation: https://onlinelibrary.wiley.com/doi/full/10.1111/j.1466-8238.2011.00694.x

Using this method of calculation, you can compare species richness and replacement between your 3 soil use types.

Line 216 you are using PAST 3.2 and line 234: PAST 4.11. Knowing that the last version is 4.12 (ok since March 2023, so the 4.11 is ok).

Line 246: Are you sure it is J. genoveva (which I think feeds exclusively on mangrove verbenaceae). Junonia are really difficult and the differences with J. evarete (Line251) are not so obvious. I think that at 20 km from the coast, the chance to find genoveva is very low.

Line 282: Please delete the subspecies for februa (as well as calydonia). In this area, the most likely is the presence of only M. helenor peleides (not two subspecies of helenor as I understand from your text). So just leave M. helenor (better in my mind) or if you consider peleides to be a separate species M. peleides.

Line 293: what a pity that you didn’t ask the help of Andy (Warren) or his student Riley Gott to check Hesperiidae as these Lepidoptera represent around half of the species.

Line 295: please check with Tisiri who is in charge of your paper for MDPI if you may put names. I think that it is just numbers under brackets. Same thing line 307/308.

Line 310: remove names also

Line 328: Please modulate your conclusion. First of all, it is much more difficult to collect in the forest and (as you have already mentioned) Lycaenidae are much more abundant in the forest but extremely difficult to collect as many species fly above the trees.

Knowing these tropical dry forests well, it is very clear that many more individuals and species can be observed in the semi-disturbed areas than in the deep forest. This is partly due to the presence of common generalist species (shared with disturbed areas) but mainly due to the fact that individuals are concentrated in the few suitable locations in the IDA zone, whereas they occupy the whole forest space and are therefore much more spread out.

Please ask help of a native speaker.

Author Response

(The authors gave the same response as above.)

Reviewer 3 Report

MAJOR COMMENTS:

The authors present a nice paper with an interesting question. However, I believe that the overall presentation could be substantially improved (especially the Methods section). In fact, it is not clear to the reader why these specific beta-diversity metrics were selected, and neither why so many model families were used to evaluate the change in richness and abundance associated with the distance from each plot to the least disturbed area of its transect. 

After reading the Methods a couple of times, the study design is still not fully clear to me (it should be noted that I did not have access to the figures and tables, since these were not included in the pdf sent to me by the journal). In any case, the description of the study design in the Methods should be self-explanatory.

I also noted that many relavant references were not inlcuded in the Introduction and Discussion sections. I mention some of these in my minor comments (see below):

MINOR COMMENTS:

Line 52: Replace 'The understanding of the patterns that comprise communities in disturbed sites' by 'The understanding of ecological patterns in disturbed communities' (or similar)

Line 55: It is good to cite in this sentence a study that used butterflies as model system, such as: Barlow et al. 2007. The value of primary, secondary and plantation forests for fruit-feeding butterflies in the Brazilian Amazon.

Line 58: Replace 'increasing the environmental filter' by 'increasing the effect of the environmental filter on species composition'

Lines 60-62: I'd replace this by: 'If areas closest to the forest are less disturbed, and the degree of disturbance increases with distance from the forest, intermediate areas should have a moderate disturbance when compared to the two extremes of the disturbance gradient.

Line 64: fundamental concepts IN ECOLOGY

Line 68: add: ...across different taxa [add more examples/refs in here]

Lines 70-77: These are important references for this paragraph:

Barlow et al. 2007. The value of primary, secondary and plantation forests for fruit-feeding butterflies in the Brazilian Amazon

Uehara-Prado et al. 2006. Species richness, composition and abundance of fruit-feeding butterflies in the Brazilian Atlantic Forest: comparison between a fragmented and a continuous landscape

Pereira Martins et al. 2017. SPECIES DIVERSITY AND COMMUNITY STRUCTURE OF FRUIT-FEEDING BUTTERFLIES (LEPIDOPTERA: NYMPHALIDAE) IN AN EASTERN AMAZONIAN FOREST

Lines 78-79: Replace by "Butterflies are an importante model group to understand how human actions have influenced the community structure of tropical dry forest biota, since they are highly sensitive to ecosystem changes.

Line 109: Replace: 'Under this approach, the following questions want to be answered" by "More specifically, we aimed to answer the following questions"

Lines 111-113: Divide those into two different questions

Line 114: Replace "been shaped to the" by "been shaped by"

Lines 133-137: You mentioned 3 linear transects, 16 circular plots and 48 plots. I am guessing Figure 1 shows your study design (I dont have access to this figure), but just reading the text is very difficult to understand exactly how your plots/traps were distributed.

Line 139: You should cite the original reference for Rydon traps (Rydon, A.H.B. 1964. Notes on the use of butterfly traps in East Africa. Journal of the Lepidopterists’ Society, 18:51-58)

Line 149: It'd be good to have references for this sentence. For example, these two studies suggest that Hesperiidae richness is often underestimated:

Butterflies of Amazon and Cerrado remnants of Maranhão, Northeast Brazil (2017) Biota Neotropica;

Butterflies (Lepidoptera, Papilionoidea and Hesperioidea) of the “Baixada Santista” region, coastal São Paulo, southeastern Brazil. Rev Bras Entomol.

Line 150: replace 'muddy' by 'blur'. 

Line 169: replace 'whom' by 'which'

Lines 184-189: Why didn't you use a more direct measure, such as land cover (i.e., distance to a forest patch/plantation)? Fror your description it seems like watersheds are regions with highest forest cover, so why didn't just use land cover as a metric? 

Lines 192-194: it is not clear why so many models were performed. Did you test all these to evaluate which one better fit your data? If yes, I'd just mention here the model that you actually used in the analysis.

Lines 201-202: I cannot see Figure 2, so its not clear why nine ranges were used. It seems that you chose the ranges from 250m to 250m; why did you use this specific distance value?

Lines 205-206: These two metrics should be better explained. What do they measure exactly?

Lines 208-209: what you mean by this? There areas were not selected based on your study design? Or did you separate the areas based on how many species they have in common? This should be clearer in the text.

Lines 212-215: Again, it should be clearer why you used these metrics. What do they measure?

Line 415: Note that other studies have found that H. februa is characteristic of disturbed habitats (you should mention that, since this may be because these studies were conducted in other biomes/habitat types):

BROWN JR, K.S. 1992. Borboletas da Serra do Japi: diversidade, habitats, recursos alimentares e variação temporal. In Historia Natural da Serra do Japi: ecologia e preservação de uma área florestal no sudeste do Brasil (L. Morellato, ed). Unicamp, Campinas, p.142–187.

Butterflies of Amazon and Cerrado remnants of Maranhão, Northeast Brazil (2017)

The overall English is good and understandable. However, some sentences were not fully clear (I describe those in my minor comments). 

Author Response

(The authors gave the same response as above.)

Round 2

Reviewer 2 Report

Dear authors

Your article is much improved, there are still parts where the English needs to be improved but the Diversity editorial office will be able to help you.

In your Table 1, for the purposes of consistency, I think the names of the subspecies should be removed. For Junonia, Nick Grishin has very recently tried to clean things up a bit (see the Butterfliesofamerica.com website) but the state of the situation remains very confused. So, either you clearly have two species and in this case put Junonia sp1 and Junonia sp2 or things are not clear (existence of genoveva outside its biotope?) and in this case pool all the Junonia together.
As it was published in Diversity and deals with the specificity of butterfly communities in tropical dry forests, this article could be cited: Legal L, Valet M, Dorado O, Jesus-Almonte JM, Lopez K, Céréghino R (2020) Lepidoptera are relevant bioindicators of passive regeneration in tropical dry forests. Diversity-Basel. 12:231
Congratulations on your efforts, your article is now much closer to being publishable.

May be a last check with the help of editorial office.

Author Response

In your Table 1, for the purposes of consistency, I think the names of the subspecies should be removed. For Junonia, Nick Grishin has very recently tried to clean things up a bit (see the Butterfliesofamerica.com website) but the state of the situation remains very confused. So, either you clearly have two species and in this case put Junonia sp1 and Junonia sp2 or things are not clear (existence of genoveva outside its biotope?) and in this case pool all the Junonia together. As it was published in Diversity and deals with the specificity of butterfly communities in tropical dry forests, this article could be cited: Legal L, Valet M, Dorado O, Jesus-Almonte JM, Lopez K, Céréghino R (2020) Lepidoptera are relevant bioindicators of passive regeneration in tropical dry forests. Diversity-Basel. 12:231 Congratulations on your efforts, your article is now much closer to being publishable.

R:// We are agree and we change the species names. 

Reviewer 3 Report

The structure of the manuscript was substantially improved and the methods/results are much clearer now when compared to the previous version of the paper. The figures the authors added also contribute a lot to make the whole manuscript easier to follow by an average reader. My only concerns now are related to typos, mispellings and the overall quality of english.

As mentioned in my above comment, the text should be revised throughout. I detected several typos, mispellings and present/past tense errors.

I particularly don't like to mention this, being a non-native english speaker myself, but unfortunaly the english language has to be substantially improved before publication. 

Author Response

As mentioned in my above comment, the text should be revised throughout. I detected several typos, mispellings and present/past tense errors.

R:// We are agree, we made some changes and a native English speaker review the manuscript. However we also hope that editorial teem of the journal should help us with the English revision. Such as suggest the Reviewer 2.
